# Changes in Diversity and Structure of Thrips (Thysanoptera) Assemblages in the Spruce Forest Stands of High Tatra Mts. after a Windthrow Calamity

**DOI:** 10.3390/insects13080670

**Published:** 2022-07-25

**Authors:** Rudolf Masarovič, Martina Zvaríková, Milan Zvarík, Oto Majzlan, Pavol Prokop, Peter Fedor

**Affiliations:** 1Department of Environmental Ecology and Landscape Management, Faculty of Natural Sciences, Comenius University, Ilkovičova 6, 842 15 Bratislava, Slovakia; rudolf.masarovic@uniba.sk (R.M.); oto.majzlan@uniba.sk (O.M.); pavol.prokop@uniba.sk (P.P.); peter.fedor@uniba.sk (P.F.); 2Department of Nuclear Physics and Biophysics, Faculty of Mathematics, Physics and Informatics, Comenius University, Mlynská Dolina, 842 48 Bratislava, Slovakia; milan.zvarik@uniba.sk; 3Institute of Zoology, Slovak Academy of Sciences, Dúbravská Cesta 9, 845 06 Bratislava, Slovakia

**Keywords:** disturbance, fire, spruce forests, thrips assemblages, Thysanoptera, windthrow

## Abstract

**Simple Summary:**

The most important disturbances in European forests are windstorms and fire. In 2004, 12,000 ha of High Tatra Mts. (Slovakia) forests were seriously damaged by strong winds. Furthermore, in 2005 the area of 250 ha of not cleared forest from fallen wood biomass after windstorms was burned down. Our study brings an overview on the influences of irregular wind and fire disturbances and subsequently human activities upon the structure of thrips communities there, where eight study plots with different after-calamity management were chosen. There were two study plots that were affected by wind and subsequent management with the extraction of fallen wood biomass and influenced by fire that were characterized by low species richness. In comparison, not cleared habitats from fallen wood biomass with significantly higher species richness of thrips and higher values of diversity were characterized mainly by the presence of grass-living species *Chirothrips manicatus* and *Limothrips denticornis*. The meadow communities with longer development and higher values of diversity were inhabited by the species-rich community mainly with *Thrips fuscipennis*, *T*. *brevicornis*, *T*. *flavus*, and *T. tabaci*. Gradually, in the process of secondary succession these thrips assemblages will be probably replaced by the thrips of the shaded and long-term forests that are mainly composed of *Oxythrips bicolor*, *O*. *ajugae*, and *Thrips pini*.

**Abstract:**

Strong winds, fire, and subsequent forest management impact arthropod communities. We monitored the diversity and changes in the community structure of forest thrips assemblages in the context of secondary succession and anthropogenic impact. There were eight study plots that were affected to varying degrees by the mentioned disturbances that were selected in the Central European spruce (*Picea abies* (L.) Karst.) forests in Slovakia. The soil photoeclectors were used to obtain thrips in the study plots during two vegetation seasons. The thrips assemblages and their attributes were analyzed by non-metric multidimensional scaling (NMDS). The significant changes in community structure, composition, stratification, species richness, and diversity of thrips assemblages that were caused by natural- (wind) and human-induced disturbance (forestry and fire) were observed in our research. Our analyses revealed a clear relationship between different thrips assemblages and impacted environment. Moreover, our results indicate that silvicolous thrips species may be useful for indicating changes and disturbances in forest ecological systems.

## 1. Introduction

Windstorms, fire, and human activities belong to the common disturbances inducing the evolution and formation of forest ecosystems, e.g., [1,2] that can change the structure and composition of forests stands drastically [3,4,5,6]. The forests in Europe are mostly disturbed by windstorms [7], which represent the most significant disturbance causing up to 53% of total forest damage that was observed in past 150 years [8]. Fire belongs to important disturbance factors in many European forests, but mainly in the boreal zone and in the Mediterranean region [7,9]. Furthermore, European forests appearance is the result of human activities [10] that can influence the representation of guild structure in forests [11,12,13]. Intensive management is often conducted in windthrown areas to prevent bark beetle outbreaks that can lead to economic losses [14]. However, it is very likely that disturbance regimes will intensify in Central European forests of mountain regions in the near future due to climate change [8,15,16]. Such agents may influence ecosystem attributes and may affect species richness and diversity of the present communities [17]. 

On 19 November 2004, the forests on the southern slope of the High Tatra Mts. (Slovakia) were seriously damaged by strong winds; the area of 12,000 hectares of the spruce (*Picea abies*) forest was laid down [18]. Furthermore, in 2005 the area of 250 ha of not completely cleared forest after windstorms was burned down. Historical records did not show a similar calamity in High Tatras for past 90 years, suggesting that its occurrence is an exception, rather than a rule. Moreover, human-induced management of timber harvesting was conducted at a huge area of forest in the majority of the affected localities. Since the forest calamity in 2004, the oldest national park in Slovakia has become a model area for studying the impact of windstorm, fire, and management on forest ecosystems [19]. In this regard, since 2005 researchers from different countries have been studying the consequences and impacts of the wind calamity very intensively e.g., [20,21,22]. 

The influences of irregular wind and fire disturbances and the subsequent human activities upon the structure of insect communities in European spruce forests is still poorly understood. In spite of the intensive research of insect recolonization after disturbances e.g., [3,23,24,25,26,27], the knowledge of the dynamics and diversity of the sap-sucking insects in spruce forest conditions remains insufficient.

In this regard, the order Thysanoptera was used as a model taxocenose to analyze the effect of the wind, fire, and human-induced disturbances on species richness, diversity, and the composition of the sap-sucking insects in Central European spruce forests. The order Thysanoptera includes many phytophagous species [28] that are related with plant communities very closely [28,29,30]. They prefer a broad spectrum of microhabitats (e.g., bark, flowers, logs, dead wood, forest litter), play a specific role as a part of the guilds (bark-dwelling, tree-dwelling, soil-dwelling etc.) and occupy different ecological niches [28,31,32]. Moreover, some species may spread to relatively unfavorable conditions [33]. Thrips communities react very sensitively to the disturbances in the forest ecosystem that are caused by ecological and environmental impacts. Their assemblages with all their attributes (e.g., abundance, species richness, diversity, and evenness) can refer to the state and condition of the ecosystem; they may be useful for indicating changes and disturbance in forests [34,35,36,37,38]. 

The aims of this work were: (1) to find out how the windstorm calamity, fire, and forestry management could affect the species richness, species diversity, and the community structure of the thrips and (2) to emphasize the ability of the thrips assemblages to indicate changes in forest ecosystems.

## 2. Materials and Methods

### 2.1. Study Area

The High Tatra Mts. represent the largest and highest mountain range situated in Slovakia. The mountains are located at the north of the country forming the border with Poland. In November 2004, the forest on the south slope was affected by a windstorm and 12,000 ha of forest trees were laid down. A relatively small part of the windthrown area was left for natural development after the wind disturbance and the larger part was cleared. Subsequently in 2005, an area that was not completely cleared was burned by a human-caused open fire.

A total of eight study plots (EXT, FIR, NEX1, NEX2, LUK1, LUK2, LES1, and LES2) were selected in the High Tatra Mts. (Figure 1, Table 1) for studying the changes in diversity of the thrips assemblages that was caused by natural (wind) as well as human forestry management and human-caused fire. The study plot EXT was characterized by extracted wood after wind calamity that represents the traditional way of forestry management. The wood was removed and the branches and thin wood were piled. The study plot FIR was distinguished by partially extracted wood after the disaster and it was incidentally burned by a human-induced fire in 2005 before the extraction of wood biomass could be completed. The plots NEX1 and NEX2 belonged to non-extracted (not cleared) areas after the wind calamity with zero management, where the fallen trees were left unmanaged. They differed in the percentage of the standing trees which were not damaged by the catastrophe. NEX1 and NEX2 consisted of 60% and 10% of the standing and 40% and 90% of the fallen trees, respectively. Meadow plots LUK1 and LUK2 with longer development were not affected by wind and represented 40 years old meadows that were mowed twice a year. Finally, LES1 and LES2 represented two control forest stands with minimal human impact and forestry management. They were not affected by the strong winds and fire. At each study plot, except LUK1 and LUK2, there was originally the same homogeneous plant community *Vaccinio myrtili-Piceetum* until the disaster occurred.

### 2.2. Sampling

Soil photoeclector traps (POT) (Figure 2) were used to obtain thrips specimens. Each stationary trap covered an area of 0.125 m^2^ (3 = 0.375 m^2^) of the soil surface and was used to collect thrips that were emerging from the soil. A total of three traps were installed at each study plot during two vegetation seasons (years 2007—plots EXT, FIR and 2008—plots NEX1, NEX2, LUK1, LUK2, LES1, and LES2) from April to October The material was collected in approximately two-week intervals (year 2007: 15 May, 28 May, 15 June, 28 June, 16 July, 24 July, 2 August, 21 August, 13 September, 8 October; year 2008: 23 May, 7 June, 21 June, 4 July, 19 July, 2 August, 16 August, 30 August, 14 September, 29 September). The thrips were captured in a collecting jar that was filled with commercial car antifreeze liquid (based on ethylene glycol) with distilled water as a conservation liquid. 

The obtained thrips were preserved in AGA solution. Afterwards, the thrips were mounted on miroscopic slides according to Mound and Kibby, 1998 [39]. The thrips were determined using the common keys for thrips identification [40,41,42]. 

### 2.3. Data Analysis

The thrips assemblage attributes, such as composition, structure, abundance, dominance, species richness, diversity, and evenness were evaluated. The diversity was calculated by the application of the Shannon index and reciprocal Simpson index of diversity [43,44], which were used for the evaluation of evenness of the thrips assemblages. 

The thrips assemblages were analyzed by non-metric multidimensional scaling (NMDS). Species abundances were transformed by square root transformation and subsequently by Wisconsin double standardization. Horn–Morisita dissimilarity index was applied to detect gradients. Environmental information was overlaid onto the plots using the function “envfit” and the environmental vectors were fitted onto ordination diagrams. Surfaces of environmental variables were fitted using the function “ordisurf”, which applied generalized additive models (GAM). Statistical methods were performed using R software using the vegan [45], MASS, and maptools packages. 

A total of 34 ecological and environmental variables (Table 2) were used to find the relationship between the thrips communities and their environment after windthrow. Using NMDS 17 out of 34 ecological and environmental variables were observed as statistically significant (α < 0.05) (Table 2).

## 3. Results

A total of 775 thrips were captured, belonging to 48 species (Table 3). There were four species that represented 73.5% of the captures: *Thrips tabaci* (21.2%), *T. pini* (20.9%), *Chirothrips manicatus* (17.4%), and *Aeolothrips intermedius* (14.1%). The NMDS revealed a relationship between thrips assemblages from different study plots that were affected by wind calamity and subsequent management method and anthropogenic impact (Table 2).

### 3.1. Vertical and Horizontal Stratification of Thrips

The ordination diagram indicates the changes between the study plots, where the forest study plot with the presence of trees, higher tree cover, and plant height (Figure 3, Table 2) were preferred by different thrips species. The biotopes with low vegetation were occupied mainly by the pratincolous heliophilous species (e.g., *Aeolothrips intermedius*, *Thrips tabaci*, *Thrips flavus*, *Thrips fuscipennis*, and *Thrips brevicornis*). In comparison, the conditions of Central European spruce forests of the High Tatras were preferred by typical forest thrips species (with e.g., *Oxythrips bicolor*, *Oxythrips ajugae*, *Thrips pini*, and *Thrips minutissimus*).

### 3.2. Functional Structure of Thrips Assemblages

Habitats that were affected by man (EXT, FIR) in the initial stage of secondary succession were distinguished by different species structure with the presence of mainly flower-living thrips species (e.g., *Thrips tabaci, Aeolothrips intermedius*) (Figure 4). On the contrary, the older meadows (LUK1, LUK2) were occupied by the species-rich and guild-rich communities that were characterized by Thrips *fuscipennis*, *Thrips brevicornis*, *Thrips flavus*, *Thrips tabaci*, and others with a representation of flower-living species, but also other functional groups. Moreover, the study plots with the presence of trees (LES1, LES2, NEX1) were occupied by leaf-living species on trees (e.g., *Thrips pini*, *Oxythrips ajugae*) instead of bark-dwelling species and other guilds. On the other side, not cleared study plot (NEX2), characterized by 90% of fallen trees after wind calamity was characterized by the presence of mainly grass-living species (e.g., *Chirothrips manicatus*, *Limothrips denticornis*) (Figure 5). The graph on Figure 5 also indicates differences in the representation of the guilds at the studied plots, where EXT and FIR are distinguished by low number of guilds compared to the rest of the study plots.

### 3.3. Species Richness and Diversity of Thrips Assemblages

The NMDS analyses revealed that EXT and FIR study plots were isolated from the other studied plots. The study plots EXT and FIR that were affected by wind calamity and clear-cut management with the extraction of fallen wood biomass (EXT, FIR) and influenced by human-induced fire (FIR) were characterized by low Shannon diversity index (0.53–0.66) (Figure 6) and low evenness (0.27–0.37) (Figure 7, Table 4), whereas the others (NEX1, NEX2, LUK1, LUK2, LES1, LES2) that were not influenced by human management showed a higher Shannon diversity (1.40–2.30) and evenness (0.53–0.83) (Table 4). 

In this sense, the group of species *Aeolothrips intermedius* and *Thrips tabaci* appears as an indicator of an imbalanced ecosystem in the early stages of secondary succession with lower values of species richness, species diversity, and equitability. Moreover, NMDS revealed that the study plots that were affected by wind calamity, but not by human management (NEX1, NEX2) were characterized by approximately similar attributes (species richness, diversity, evenness) to the meadow (LUK1, LUK2) and the reference forest study plots (LES1, LES2) (Figure 6 and Figure 7, Table 4). This community is characterized mainly by the presence of grass-living species (e.g., *Chirothrips manicatus*, *Limothrips denticornis*). Grass-herbaceous plant communities (LUK1, LUK2) with longer development are characterized by higher values of abundance, diversity and evenness of thrips species. These meadows are indicated by the formation of *Thrips fuscipennis*, *Thrips brevicornis*, *Thrips flavus*, and *Thrips tabaci* assemblages. Finally, long-term forest ecological systems with diverse and equally represented communities are mainly composed of forest species *Oxythrips bicolor*, *Oxythrips ajugae*, and *Thrips pini.*

### 3.4. Anthropogenic Impact 

Changes in the diversity and equitability are most probably related to anthropogenic processes in the forest ecosystems of the High Tatras. Anthropogenic impact was based on six negative points of anthropogenic activity (Table 1), when a significant degree of anthropogenic disturbance (Figure 8) was indicated mainly by the generalists and eurytopic species (e.g., *Thrips tabaci*, *Aeolothrips intermedius*), that were able to live in a wide variety of habitats and tolerate a wide range of environmental conditions. On the contrary, for example the community of forest species *Oxythrips bicolor*, *Oxythrips ajugae*, and *Thrips pini* showed a closer preference to conditions without significant anthropogenic impact. 

Our results are summarized in the scheme (Figure 9), which indicates how the structure, diversity, and composition of thrips assemblages and their attributes gradually change depending on anthropogenic disturbance and process of secondary succession. The results indicate that wind and fire, exclusively in combination with human activities have significantly changed the characteristics of thrips assemblages in the High Tatras spruce forest ecosystem. 

## 4. Discussion

Strong winds and fire push succession to its initial stages and play an important role in the successional cycle of natural forests, which influences the structure and composition of forests [3]. During this process open habitats with different microclimatic conditions and resource supply are created [3,46]. Consequently, significantly different species composition of thrips assemblages was found in the open windthrow and meadow areas compared to the reference intact forest stands. Moreover, in another study that was carried out in the High Tatras [22], it was concluded that fire disturbance had the most extensive influence on vegetation regeneration in Central European spruce forests. The burned plots had significantly different plant composition than the other studied plots. Correspondingly, our findings showed that the burned plots were colonized mainly by *Thrips tabaci* that probably represents an early successional r-selected opportunist that is associated with the pioneer plant community in the early stages of the succession. 

Furthermore, species composition, species richness, and the diversity of thrips assemblages differed between the studied plots and we have noticed an overall increase in thrips diversity in such disturbed landscape compared to only intact forest stands, which can be related to intermediate disturbance theory [47] and mosaic concept [48]. The creation of the new mosaic forests and gaps may significantly increase the species richness of invertebrates [25]. The most anthropogenically-influenced study sites are characterized by the lowest values of species richness, species diversity, and evenness. On the other hand, uncleared study plots that were influenced by wind calamity that were left for self-development had similar values of species richness compared to meadow communities and intact forest stands. Our results show that clearing windthrown spruce forests following a catastrophic storm is less favorable for arthropod communities, which is similar to the conclusions of other studies from the High Tatra mountains e.g., [49,50,51,52] but see [53]. Moreover, the uncleared study area yielded more species than the intact forest plot. Nevertheless, the highest values of Shannon index diversity and evenness have been noticed at the meadows, in all probability because meadows are occupied by more plant species than spruce forests. 

Anthropogenic impact, especially clearing treatment may cause a decrease in species richness [54,55], but see [6,56]. On the other hand, species richness and abundance of insects in gaps after windstorms can be higher than in the neighboring plots of control forests [6,25,54] and a combination of cleared and uncleared areas may enhance biodiversity significantly [6,25,56]. Thus, the flower-rich salvaged part and the coarse woody debris uncleared part and intact forests could favor biodiversity at the regions that are impacted by severe windstorms [3]. 

In general, wind, fire, and subsequent human management create the opportunity to colonize affected habitats by species that are characteristic of open habitats. The composition of thrips assemblages changed dramatically. There were two study plots that were affected by wind calamity and subsequent management with extraction of fallen wood biomass and submitted to human-induced fire that were characterized by low species richness, diversity, and evenness that were colonized mainly by *Thrips tabaci* and *Aeolothrips intermedius*, which may occur together in flowers on a variety of herbs [57,58] in similar open habitats [59,60]. These species may prefer unbalanced communities [61,62] probably with the low diversity indices. In comparison, not cleared habitats after wind calamity with significantly higher species richness of thrips and higher values of diversity are characterized mainly by the presence of grass-living species *Chirothrips manicatus* and *Limothrips denticornis*. These species are normally tolerant to a wide variety of environmental conditions and are characterized by strong dispersal mechanisms, which allow them to occupy and spread to variable environments [63,64]. The meadow communities that are characterized by longer development and higher values of diversity were inhabited by the species-rich community mainly with *Thrips fuscipennis*, *T. brevicornis*, *T. flavus*, and *T. tabaci*. Information on the preference of relatively stable assemblages by the *Thrips fuscipennis* and *T. brevicornis* is incomplete in the literature. Nevertheless, some papers indicate their preference for natural meadow communities in the mountainous regions e.g., [30,65]. The high abundance of *Thrips flavus* on these meadow habitats can be probably explained by the presence of the food plant *Chamaenerion angustifolium* [66,67]. *Oxythrips bicolor*, *O. ajugae*, and *Thrips pini* showed a closer preference to conditions without significant anthropogenic impact. *Oxythrips* species normally live in old coniferous forest stands [65,68]. Moreover, another arboricolous taxon *Thrips pini* often occurs with them, as it was in the Białowieza pine and spruce forests [68,69] and in the old natural stands in Finland [70]. Although *Thrips pini* tends to prefer more diverse and stable forests (LES1, LES2), it was also found in nearby meadows (LUK1, LUK2) and non-extracted plots (NEX1) with the presence of spruce trees during our research, which indicate its eurytopic character in the conditions of the High Tatra spruce forests. Furthermore, this species is also known as pest species from unnatural economic forests [71,72] with relatively effective dispersal mechanisms [73,74].

## Figures and Tables

**Figure 1 insects-13-00670-f001:**
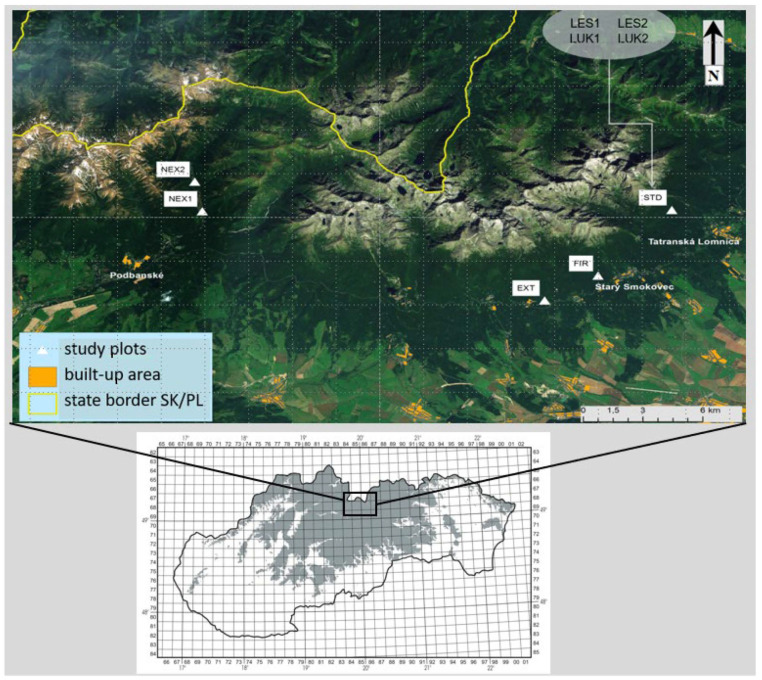
Study area with the selected study plots in the High Tatra Mountains.

**Figure 2 insects-13-00670-f002:**
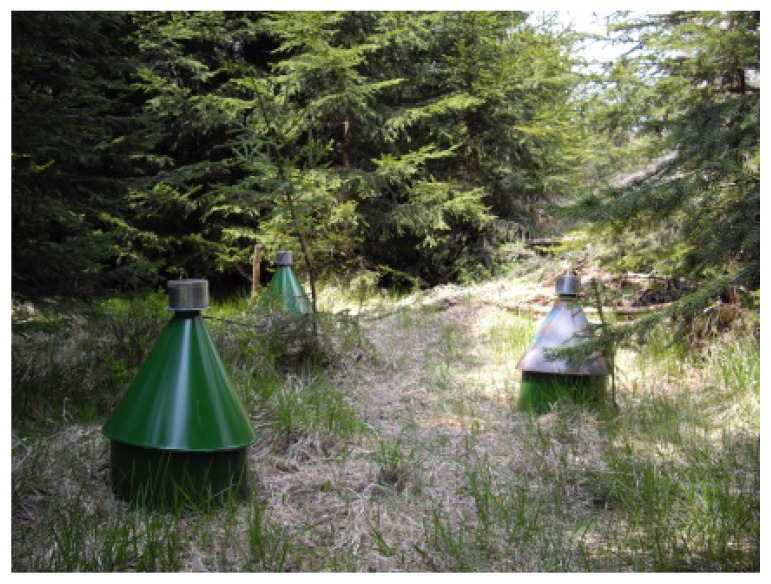
The soil photoeclectors that were used to collect thrips.

**Figure 3 insects-13-00670-f003:**
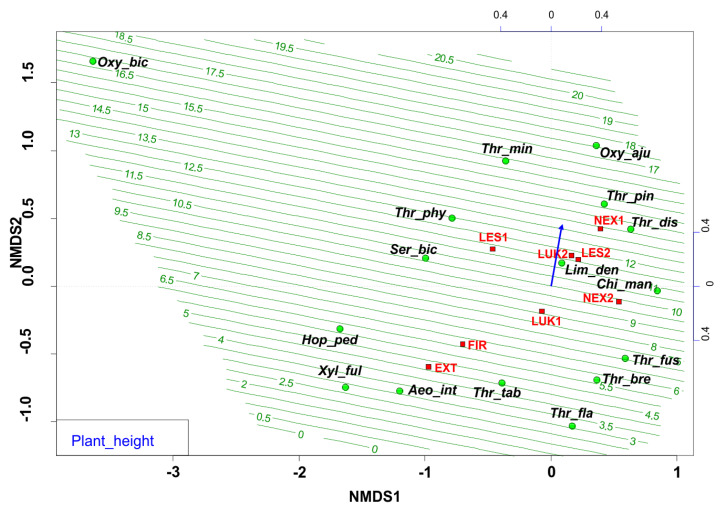
Nonmetric multidimensional scaling (NMDS) plot showing the relatedness of the thrips assemblages between the different samples and study plots (EXT, FIR, NEX1, NEX2, LUK1, LUK2, LES1, LES2) and the plant height of the present plant community. Oxy_bic–*Oxythrips bicolor*, Oxy_aju–*Oxythrips ajugae*, Thr_min–*Thrips minutissimus*, Thr_pin–*Thrips pini*, Thr_dis–*Thrips discolor*, Thr_phy–*Thrips physapus*, Ser_bic–*Sericothrips bicornis*, Lim_den–*Limothrips denticornis*, Chi_man–*Chirothrips manicatus*, Hop_ped–*Hoplothrips pedicularius*, Xyl_ful–*Xylaplothrips fuliginosus*, Aeo_int–*Aeolothrips intermedius*, Thr_tab–*Thrips tabaci*, Thr_fus–*Thrips fuscipennis*, Thr_bre–*Thrips brevicornis*, Thr_fla–*Thrips flavus*. EXT, FIR, LES1, LES2, LUK1, LUK2, NEX1, NEX2–study plots.

**Figure 4 insects-13-00670-f004:**
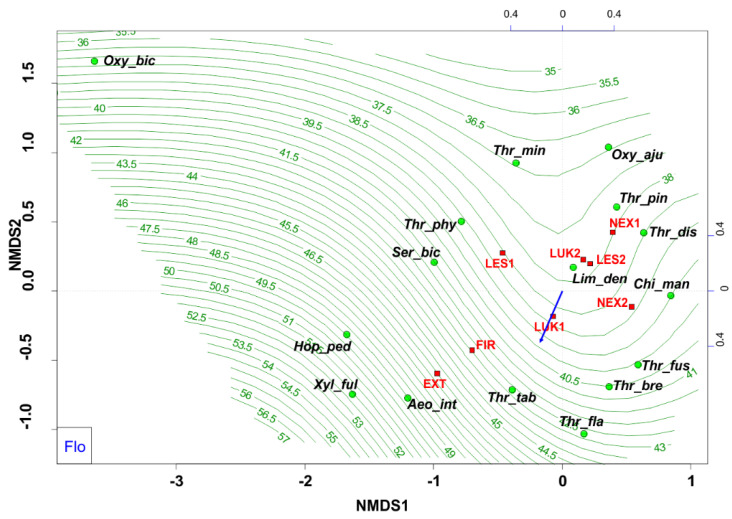
Nonmetric multidimensional scaling (NMDS) plot showing the relatedness of the thrips assemblages between the different samples and study plots (EXT, FIR, NEX1, NEX2, LUK1, LUK2, LES1, LES2) and the present of floricolous species. Oxy_bic–*Oxythrips bicolor*, Oxy_aju–*Oxythrips ajugae*, Thr_min–*Thrips minutissimus*, Thr_pin–*Thrips pini*, Thr_dis–*Thrips discolor*, Thr_phy–*Thrips physapus*, Ser_bic–*Sericothrips bicornis*, Lim_den–*Limothrips denticornis*, Chi_man–*Chirothrips manicatus*, Hop_ped–H*oplothrips pedicularius*, Xyl_ful–*Xylaplothrips fuliginosus*, Aeo_int–*Aeolothrips intermedius*, Thr_tab–*Thrips tabaci*, Thr_fus–*Thrips fuscipennis*, Thr_bre–*Thrips brevicornis*, Thr_fla–*Thrips flavus*. EXT, FIR, LES1, LES2, LUK1, LUK2, NEX1, NEX2– study plots.

**Figure 5 insects-13-00670-f005:**
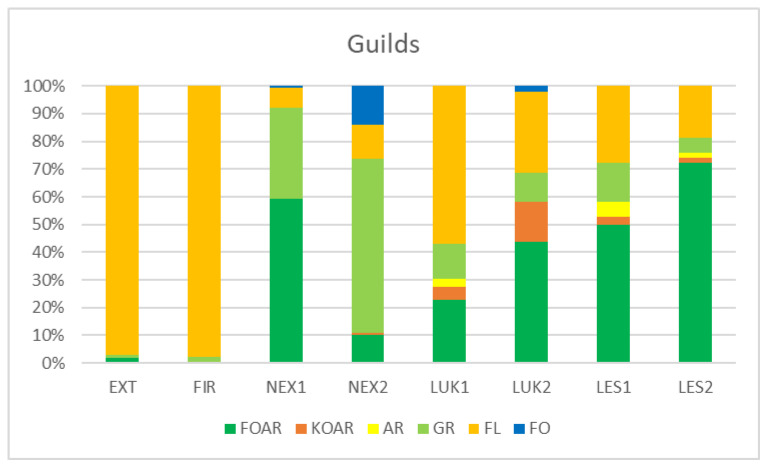
Percentage representation of species abundance belonging to the guilds at the studied plots. FOAR—foliicolous arboricolous species, KOAR—corticicolous species, AR—arboricolous species, GR—graminicolous species, FL—floricolous species, FO—Foliicolous species.

**Figure 6 insects-13-00670-f006:**
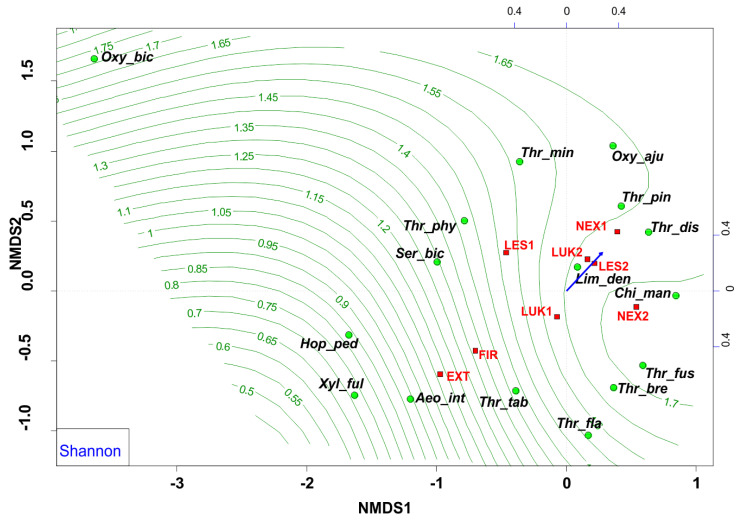
Nonmetric multidimensional scaling (NMDS) plot showing the relatedness of the thrips assemblages between the different samples and study plots (EXT, FIR, NEX1, NEX2, LUK1, LUK2, LES1, LES2) and the Shannon diversity index (blue arrow, green contour line). Oxy_bic–*Oxythrips bicolor*, Oxy_aju–*Oxythrips ajugae*, Thr_min–*Thrips minutissimus*, Thr_pin–*Thrips pini*, Thr_dis–*Thrips discolor*, Thr_phy–*Thrips physapus*, Ser_bic–*Sericothrips bicornis*, Lim_den–*Limothrips denticornis*, Chi_man–*Chirothrips manicatus*, Hop_ped–*Hoplothrips pedicularius*, Xyl_ful–*Xylaplothrips*
*fuliginosus*, Aeo_int–*Aeolothrips intermedius*, Thr_tab–*Thrips tabaci*, Thr_fus–*Thrips fuscipennis*, Thr_bre–*Thrips brevicornis*, Thr_fla–*Thrips flavus*. EXT, FIR, LES1, LES2, LUK1, LUK2, NEX1, NEX–study plots.

**Figure 7 insects-13-00670-f007:**
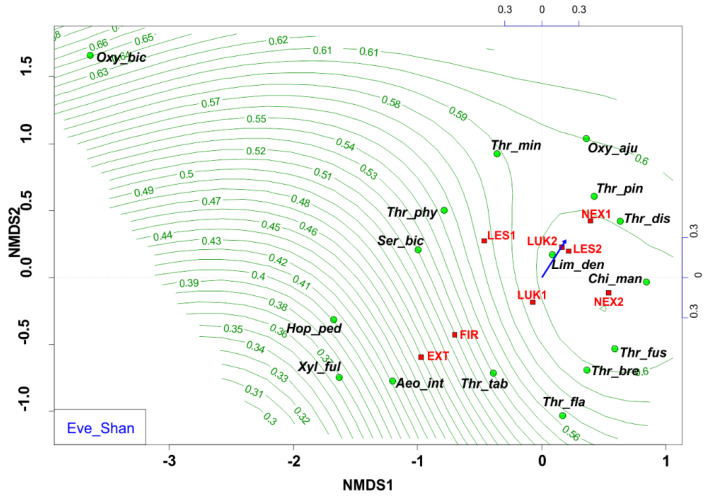
Nonmetric multidimensional scaling (NMDS) plot showing the relatedness of the thrips assemblages between the different samples and study plots (EXT, FIR, NEX1, NEX2, LUK1, LUK2, LES1, LES2) and the Shannon evenness (blue arrow, green contour line). Oxy_bic–*Oxythrips bicolor*, Oxy_aju–*Oxythrips ajugae*, Thr_min–*Thrips minutissimus*, Thr_pin–*Thrips pini*, Thr_dis–*Thrips discolor*, Thr_phy–*Thrips physapus*, Ser_bic–*Sericothrips bicornis*, Lim_den–*Limothrips denticornis*, Chi_man–*Chirothrips manicatus*, Hop_ped–*Hoplothrips pedicularius*, Xyl_ful–*Xylaplothrips fuliginosus*, Aeo_int–*Aeolothrips intermedius*, Thr_tab–*Thrips tabaci*, Thr_fus–*Thrips fuscipennis*, Thr_bre–*Thrips brevicornis*, Thr_fla–*Thrips flavus*. EXT, FIR, LES1, LES2, LUK1, LUK2, NEX1, NEX–study plots.

**Figure 8 insects-13-00670-f008:**
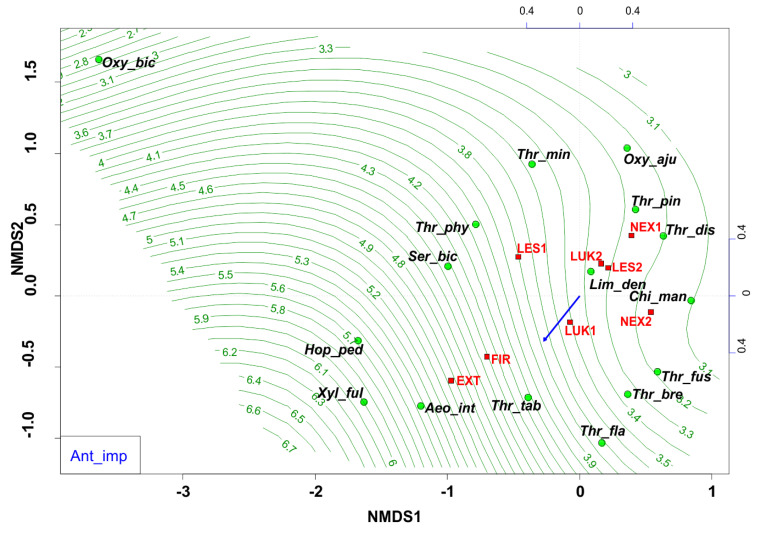
Nonmetric multidimensional scaling (NMDS) plot showing the relatedness of the thrips assemblages between the different samples and study plots (EXT, FIR, NEX1, NEX2, LUK1, LUK2, LES1, LES2) and the impact of anthropogenic impact (blue arrow, green contour line). Oxy_bic–*Oxythrips bicolor*, Oxy_aju–*Oxythrips ajugae*, Thr_min–*Thrips minutissimus*, Thr_pin–*Thrips pini*, Thr_dis–*Thrips discolor*, Thr_phy–*Thrips physapus*, Ser_bic–*Sericothrips bicornis*, Lim_den–*Limothrips denticornis*, Chi_man–*Chirothrips manicatus*, Hop_ped–*Hoplothrips pedicularius*, Xyl_ful–*Xylaplothrips fuliginosus*, Aeo_int–*Aeolothrips intermedius*, Thr_tab–*Thrips tabaci*, Thr_fus–*Thrips fuscipennis*, Thr_bre–*Thrips brevicornis*, Thr_fla–*Thrips flavus*. EXT, FIR, LES1, LES2, LUK1, LUK2, NEX1, NEX–study plots.

**Figure 9 insects-13-00670-f009:**
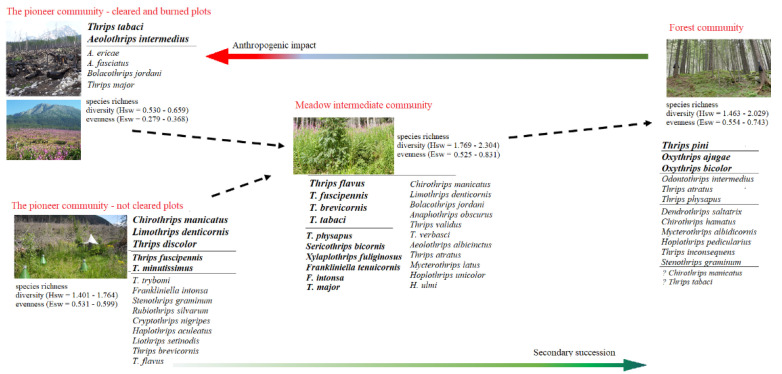
The changes in thrips assemblages in the context of secondary succession and anthropogenic impact in Central European spruce forests after a windthrow calamity.

**Table 1 insects-13-00670-t001:** Characterization of the study plots.

	EXT	FIR	NEX1	NEX2	LUK1	LUK2	LES1	LES2
Coordinates	49°07′15.60″ N	49°08′10.86″ N	49°09′41.34″ N	49°10′40.20″ N	49°10′30.02″ N	49°10′27.08″ N	49°10′29.56″ N	49°10′30.51″ N
	20°09′49.68″ E	20°11′57.00″ E	19°55′26.88″ E	19°55′02.34″ E	20°14′48.64″ E	20°14′49.52″ E	20°14′46.81″ E	20°14′49.64″ E
Altitude (m.a.s.l.)	1060	1080	1006	1052	1050	1050	1050	1050
Area (ha)	93	90	11	21.4	0.2	0.2	66	1
Age	4	4	5	5	40	40	140	140
Management	cleared	fire + cleared	not cleared	not cleared	mow	mow	no	no
Fallen trees	100%	100%	40%	90%	0%	0%	0%	0%
	Affected by windstorm				Not affected by windstorm			
Wood decay	1	1	0	0	1	1	0	0
Same-age trees	1	1	1	1	0	0	0	0
Management	1	1	0	0	1	1	0	0
Spruce %	1	1	1	1	0	0	0	0
Fire	0	1	0	0	0	0	0	0
Conservation	1	1	1	1	1	1	0	1
Anthropogenic impact	5	6	3	3	3	3	0	1

EXT, FIR, NEX1, NEX2, LUK1, LUK2, LES1, and LES2—study plots, age (years), fallen trees (percentage of fallen trees caused by wind calamity), wood decay (0-present/1-absent), same-age trees (1-present/0-absent), management: human-induced management (1-present/0-absent), spruce %: high percentage of spruces (*Picea abies*) in forest stands, fire: human-induced fire (1-present/0-absent), conservation: significantly protected areas (0-higher degree of protection)—human forestry management prohibited.

**Table 2 insects-13-00670-t002:** Description of the analyzed variables and statistical significance.

Variable	Description	NMDS1	NMDS2	r^2^	Pr(>r)	Stress	Code
Diversity							
Abundance	the number of thrips specimen per study plot	0.9378	−0.3473	0.0401	0.180	0.1271	
Eve_Shan	equitability based on the Shannon index	0.5661	0.8243	0.1125	0.008	0.1271	**
Eve_Simp	equitability based on the Simpson index	−0.9558	0.2939	0.0441	0.149	0.12722	
Shannon	Shannon–Wiener index of diversity	0.7051	0.7091	0.1492	0.002	0.1272	**
Simpson	a reciprocal Simpson diversity	0.5362	0.8441	0.0330	0.245	0.1271	
Spe_rich	the number of thrips species per study plot	0.8963	0.4435	0.2559	0.001	0.1273	***
Guild ecology							
Arb	percentage representation of species belonging to the tree-living species	−0.6883	0.7254	0.0495	0.118	0.1271	
Cor_arb	percentage representation of species belonging to the bark-dwelling species	0.7861	0.6181	0.0065	0.757	0.1271	
Flo	percentage representation of species belonging to the flower-living species	−0.4184	−0.9083	0.1618	0.001	0.1282	***
Fol	percentage representation of species belonging to the leaf-living species	0.8700	0.4930	0.0611	0.073	0.1271	
Fol_arb	percentage representation of species belonging to the leaf-living species on trees	0.6562	0.7546	0.1446	0.003	0.1273	**
Gra	percentage representation of species belonging to the grass-living species	−0.8320	0.5547	0.0084	0.690	0.1274	
Nm_guilds	the number of guilds per study plot	0.6213	0.7836	0.0285	0.299	0.1280	
Stand variables and anthropogenic impact							
Age	the age of the stand expressed in years	−0.4476	0.8942	0.0457	0.139	0.1277	
Area	area of study plot (in hectares)	−0.5999	−0.8000	0.3327	0.001	0.1272	***
Calamity	the presence of windthrow (1-present, 0-absent)	0.5430	−0.8398	0.0217	0.407	0.1279	
Defoliation_NFC	the degree of defoliation according to National Forestry Centre	0.7180	−0.6960	0.0118	0.617	0.1271	
E1	the cover of herbs	0.3671	−0.9302	0.0498	0.116	0.1270	
E2	the cover of shrubs	0.8569	0.5154	0.3691	0.001	0.1274	***
E3	the cover of trees	−0.0790	0.9969	0.1067	0.010	0.1272	**
Edge_effect	distance (in meters) of the trap from the neighboring habitat	−0.6497	−0.7602	0.3879	0.001	0.1273	***
Exposure	exposure to the sun	−0.8814	−0.4724	0.3491	0.001	0.1272	***
Health_NFC	the degree of forest health according to National Forestry Centrum	−0.1813	−0.9834	0.0561	0.089	0.1275	
Heterogeneity	habitat heterogeneity (0-weak to 1-greater)	0.6704	0.7420	0.3975	0.001	0.1276	***
Plant_com	the type of the plant community	−0.4693	0.8831	0.0454	0.140	0.1273	
Plant_height	the height of the stand in the study plot	0.1812	0.9835	0.2013	0.001	0.1274	***
Slope	slope of the study plot (in degrees)	−0.9988	0.0488	0.1221	0.004	0.1271	**
Spe_rich_tree	the number of tree species per study plot	−0.6485	−0.7612	0.0566	0.087	0.1274	
Tree_thick	the thickness of trees at the study plot	0.1374	0.9905	0.1823	0.001	0.1280	***
Ant_imp	anthropogenic impact based on six negative points of anthropogenic activity	−0.6534	−0.7570	0.1718	0.001	0.1270	***
Climatic factors							
C	clouds amount per study plot	0.9972	0.0747	0.0253	0.329	0.1273	
H	average relative air humidity during the vegetation season per study plot	0.5979	0.8016	0.2723	0.001	0.1272	***
P	the average precipitation during the growing season per study plot	0.8202	−0.5721	0.0142	0.548	0.1273	
T	average air temperature during the vegetation season per study plot	−0.8266	−0.5628	0.3532	0.001	0.1275	***

Significance levels: ** *p* < 0.01, *** *p* < 0.001.

**Table 3 insects-13-00670-t003:** The thrips species composition that was recorded on the individual study plots.

Species/Study Plot	EXT	FIR	NEX1	NEX2	LUK1	LUK2	LES1	LES2
*Aeolothrips albicinctus*					+			
*Aeolothrips ericae*	+							
*Aeolothrips intermedius*	+	+		+	+			
*Aeolothrips vittatus*								+
*Anaphothrips obscurus*		+			+			
*Aptinothrips stylifer*								+
*Bolacothrips jordani*	+	+				+		
*Dendrothrips saltatrix*							+	
*Frankliniella intonsa*		+		+				
*Frankliniella tenuicornis*				+		+		
*Chirothrips hamatus*							+	
*Chirothrips manicatus*		+	+	+	+	+	+	+
*Limothrips denticornis*			+	+	+		+	
*Mycterothrips albidicornis*							+	
*Mycterothrips latus*					+			
*Neohydatothrips gracilicornis*			+					
*Odontothrips intermedius*							+	
*Odontothrips loti*			+	+				
*Oxythrips ajugae*			+	+	+	+		+
*Oxythrips bicolor*					+			+
*Rubiothrips silvarum*			+					
*Sericothrips bicornis*					+		+	
*Stenothrips graminum*			+					+
*Taeniothrips inconsequens*							+	
*Taeniothrips picipes*			+	+				
*Thrips angusticeps*						+		+
*Thrips atratus*							+	+
*Thrips brevicornis*				+	+	+		
*Thrips discolor*			+	+				
*Thrips flavus*			+	+	+	+	+	+
*Thrips fuscipennis*			+	+	+	+		
*Thrips major*		+				+		
*Thrips minutissimus*	+		+	+	+			+
*Thrips physapus*				+	+		+	+
*Thrips pini*			+	+	+	+	+	+
*Thrips praetermissus*						+		
*Thrips tabaci*	+	+	+	+	+	+	+	+
*Thrips trybomi*				+				
*Thrips validus*					+			
*Thrips vulgatissimus*								+
*Cryptothrips nigripes*				+				
*Haplothrips aculeatus*				+				
*Hoplothrips pedicularius*					+	+	+	+
*Hoplothrips polystici*						+		
*Hoplothrips ulmi*					+			
*Hoplothrips unicolor*						+		
*Liothrips setinodis*				+				
*Xylaplothrips fuliginosus*					+	+		

**Table 4 insects-13-00670-t004:** Diversity attributes of the study plots.

	EXT	FIR	NEX1	NEX2	LUK1	LUK2	LES1	LES2
Spe_rich	6	7	14	20	20	16	14	15
Shannon	0.6584	0.5300	1.4008	1.7642	2.3046	2.3042	1.9610	1.4630
Simpson	1.4146	1.3179	2.7297	2.8810	6.4968	7.1111	4.1538	2.3180
Eve_Shannon	0.3675	0.2724	0.5308	0.5889	0.7693	0.8311	0.7431	0.5544
Eve_Simpson	0.2358	0.1883	0.1950	0.1441	0.3248	0.4444	0.2967	0.1656

Spe_rich: species richness, Shannon: Shannon–Wiener diversity index, Simpson: reciprocal Simpson diversity index, Eve_Shannon: “Shannon evenness”, Eve_Simpson: “Simpson evenness”.

## Data Availability

The data presented in the study are available on request from corresponding author. The data are not publicly available due to the particular nature of conservation policy.

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
