# Peer review of "Changes in Diversity and Structure of Thrips (Thysanoptera) Assemblages in the Spruce Forest Stands of High Tatra Mts. after a Windthrow Calamity"

_insects, 2022, doi:10.3390/insects13080670_

Round 1

Reviewer 1 Report

This manuscript refers to the effect of strong winds, fire and human forest management in thrips assemblages in the spruce forests in the High Tatra Mountains. The objective and methodology are well explained and results adequately analysed. However discussion could be condensed/ more focused, avoiding repetition of ideas and concepts between paragraphs. Authors should update bibliographic references to support/ confront their data.

Line 14- “unextracted forest” or” non-extracted” (line 19) are not sufficiently informative of what the authors mean, especially in the beginning of the manuscript. 

Only in line 225 the authors explain what they mean: “The study plots EXT and FIR that were affected by wind calamity and clear-cut management with extraction of fallen wood biomass”

Throughout the manuscript the authors should consider to change these expressions to “not cleared” (not cleared forest), which is also used in the manuscript. But al least the concept of “extracted” must be explained.

Line 14- Suggestion: “…250 ha of not cleared forest from fallen wood biomass”

Line 23- T. flavus (also in line 367)

Line 30- “spruce” should be followed by the scientific name of the tree species.

Line 45- All references must be numbered, according to the journal instructions

Line 51- in the Mediterranean region. This is a constant in all the manuscript: “the”, “a”, “an” is often missing. Some examples are given below, but the authors should review all the manuscript for this.

Line 63- Again, “spruce” should be followed by the scientific name of the tree

Line 64- a similar

Line 65- is an exception

Line 78 and 81- sap-sucking insects

Line 81- They prefer a broad

Line 94 and 96- (1), (2)

Line 103- A relatively small…..

Along the manuscript there are several examples of words that are not correctly cut between two consecutive lines, as occurs here with “devel-opment”.

Line 105- Not clear: it is an area that was partially cleared from the fallen wood? Part of the fallen wood was removed? The same doubt in Line 110, when the authors explain FIR plot.

Line 106-125- There is no reference to the area of the plots studied.

Lines 106 and 109- You should use “study site” or “study plot” and not use both in the same manuscript

Line 120- Figure 1. Study area with selected study plots in the High Tatra Mountains

Line 124-  Characterization of the study plots.

Line 125- Instead of “anthropic”, it is better “anthropo”, and then explain in the legend. Or “human impact”.

Line 127- percentage of fallen trees

Line 129- in the legend “spruce %”, but in the table “spruce dominat”

Line 134: covered an area

Line 139-  The soil photoeclectors used to collect thrips

Line 152-160- attention to the font (different from the rest of the page)

Line 163-164 - NMDS, 17 out of 34 ecological…variables were statistically significant (α< 0.05)

Line 166- a total of 757 thrips were captured, belonging to 31 species.  Four species represented 75% of the captures: Thrips tabaci…

Line 197- Description of the analysed variables and statistical significance

In table 2- attention to “abundance”. “Study site” means “study plot”?

Line 204- species (e.g. Aeolothrips…)

Line 207- forests (e.g. Oxythrips….)

Line 224- ….isolated from the other

Table 3- maintain the order of study sites: LUK1, LUK 2, LES1, LES2

Line 279- Moreover, in another research…it was concluded that….

Line 302- favour insect species diversity

Line 304- anthropogenically

Line 324- harvesting?

Line 349-351- sentence not clear.

Line 353- …submitted to human induced fire…

Line 359- …diversity, are characterized…

Fig 9- attention to the printing quality of this figure- some of the species names are not well visible.

Author Response

Dear reviewer,

thank you very much for your comments and corrections. We do appreciate and accept all your notes, whilst many discrepancies were revealed what did improve the study. Please find our reactions to your notes below in attachment.

Authors

Reviewer 2 Report

The manuscript reports a case study about the effect of natural and anthropogenic disturbance on thrips assemblages. It reports a dramatic change in the composition of thrips assemblages between undisturbed and disturb habitats, which indicates that silvicolous thrips species are useful for indicating disturbances in forest ecological systems. In general, the language of the text only needs minor improvement. I indicated a few corrections in the ms.

Please compare your thrips catches with the catches of some other studies using the same trap type. 757 thrips individuals in this two-year study would mean an average of 1.3 thrips catches in a single trap for a two weeks period. This seems to be a fairly low figure for me. Do you consider this sample size large enough for the purpose of this paper?

To what extent do you think the trapping method influenced the representation of the different guilds in the samples? It should be discussed in the text.

Author Response

Dear reviewer,

thank you very much for your detailed work on our study. We appreciate all the comments and suggestion you gave us. They indispuptably improve the study. Please find our reactions to your notes below in the attachment. 

Authors

Reviewer 3 Report

Remarques for authors which I have included as comments on the manuscript:

I. correct the mistakes in writing the names of Limothrips denticornis and Thrips flavus

 II. Material and methods: 2.1.Study area and the study plots coordinates (mainly altitude above sea level)

2.2. Sampling – add the information on what kind of conservation medium was used in the traps

III. Results it would be interesting and useful for other scientists to know the thrips species composition on all 8 sites, not only analysis of dominant species. I suggest adding one table more with the complete list of found species.

3.3. Did you find Aptinothrips stylifer or A. rufus (in this article there is no information about these species) – Pelikán and Kucharczyk et. al, found the differences in the abundance of both of the species according to the altitude

 IV. Discussion

In discussion, the authors may answer my question formulated above (presence of Aptinothrops ssp.)

 References

Please adjust the way of citing the literature to the required rules in the “Insects” journal. Currently, some journals have a full name and others have an abbreviated version, some in italics, others not

Author Response

Dear reviewer,

thank you so much for all your corrections, notes and suggestions. We appreciate and accept all of them. It did help us a lot. Please, find attached our reactions to your notes below in the attachment.

Authors 
